# Repurposing Avermectins and Milbemycins against *Mycobacteroides abscessus* and Other Nontuberculous Mycobacteria

**DOI:** 10.3390/antibiotics10040381

**Published:** 2021-04-03

**Authors:** Lara Muñoz-Muñoz, Carolyn Shoen, Gaye Sweet, Asunción Vitoria, Tim J. Bull, Michael Cynamon, Charles J. Thompson, Santiago Ramón-García

**Affiliations:** 1Department of Microbiology, Faculty of Medicine, University of Zaragoza, 50009 Zaragoza, Spain; avitagreda@gmail.com; 2Microbiology Unit, Clinical University Hospital Lozano Blesa, 50009 Zaragoza, Spain; 3State University of New York Upstate Medical Center, Syracuse, NY 13210, USA; shoenc@cnyrc.org (C.S.); Michael.Cynamon@va.gov (M.C.); 4Department of Microbiology and Immunology, Centre for Tuberculosis Research, Life Sciences Institute, University of British Columbia, Vancouver, BC V6T 1Z3, Canada; gsweet@mail.ubc.ca (G.S.); cthompso@mail.ubc.ca (C.J.T.); 5Institute for Infection & Immunity, St. George’s University of London, London SW17 0RE, UK; tbull@sgul.ac.uk; 6Research & Development Agency of Aragón (ARAID) Foundation, 50018 Zaragoza, Spain; 7CIBER Enfermedades Respiratorias (CIBERES), Instituto de Salud Carlos III, 28029 Madrid, Spain

**Keywords:** avermectins, nontuberculous mycobacteria, *Mycobacteroides abscessus*, selamectin, milbemycin oxime, repurposing

## Abstract

Infections caused by nontuberculous mycobacteria (NTM) are increasing worldwide, resulting in a new global health concern. NTM treatment is complex and requires combinations of several drugs for lengthy periods. In spite of this, NTM disease is often associated with poor treatment outcomes. The anti-parasitic family of macrocyclic lactones (ML) (divided in two subfamilies: avermectins and milbemycins) was previously described as having activity against mycobacteria, including *Mycobacterium tuberculosis*, *Mycobacterium ulcerans,* and *Mycobacterium marinum*, among others. Here, we aimed to characterize the in vitro anti-mycobacterial activity of ML against a wide range of NTM species, including *Mycobacteroides abscessus*. For this, Minimum Inhibitory Concentration (MIC) values of eight ML were determined against 80 strains belonging to nine different NTM species. Macrocyclic lactones showed variable ranges of anti-mycobacterial activity that were compound and species-dependent. Milbemycin oxime was the most active compound, displaying broad-spectrum activity with MIC lower than 8 mg/L. Time kill assays confirmed MIC data and showed bactericidal and sterilizing activity of some compounds. Macrocyclic lactones are available in many formulations and have been extensively used in veterinary and human medicine with suitable pharmacokinetics and safety properties. This information could be exploited to explore repurposing of anti-helminthics for NTM therapy.

## 1. Introduction

The incidence of documented infections caused by nontuberculous mycobacteria (NTM) are on the rise worldwide and becoming a new neglected global health concern [1,2,3]. Reasons for this include NTM outbreaks in cosmetic and surgical procedures, potential transmission of *Mycobacteroides abscessus* among patients with cystic fibrosis (CF), and the increasing number of vulnerable individuals at risk of developing these infections, i.e., immunocompromised patients or patients with chronic pulmonary pathologies [4,5,6]. The true global burden of NTM disease is vastly under-diagnosed due to several unresolved obstacles, including the non-specific presentation of NTM disease and limited diagnostic capacity in many countries, leading to under diagnosis [1,2,3].

NTM are an extremely diverse group, with more than 190 species currently identified and new species being frequently reported [7,8]. Most common diseases caused by NTM are pulmonary, disseminated, or skin and soft tissues infections [9,10]. Pulmonary NTM infections are typically caused by *Mycobacterium avium-intracellulare complex* (MAC), *Mycobacterium kansasii,* and *M. abscessus* complex (MABSC) [11,12]. Disseminated infections can be found in immunocompromised patients, including those that have received an organ transplant or have HIV-infection. Skin and soft tissue infections are often associated with trauma, surgical procedures, or contaminated medical equipment with MABSC, *Mycobacterium chelonae*, *Mycobacterium fortuitum*, *Mycobacterium marinum*, and *Mycobacterium ulcerans* as the most prevalent pathogenic NTM [4,11].

NTM are intrinsically resistant to most antibiotics used in the clinic and patients infected with NTM require combination treatments with a minimum of 2–3 antibiotics for several months (at least 12 months therapy for respiratory or disseminated) and 4–6 months for skin infections [2,8,9,10]. Such prolonged regimens are difficult to tolerate and compliance is challenging [3,8,13]. In spite of these aggressive and lengthy treatments, cure rates are low: only 50 to 88% of MAC patients and 25 to 58% of MABSC patients achieve sputum culture conversion in respiratory disease [2]. There is an urgent need to identify new compounds that could be used against infections caused by NTM to make therapy more effective and reduce its duration [13].

Macrocyclic lactones (ML) are a family of known anti-parasitics divided into two subfamilies: avermectins and milbemycins. Avermectins are produced by *Streptomyces avermitilis* and include: ivermectin, abamectin, emamectin, selamectin, doramectin, and eprinomectin. Milbemycins, derived from metabolites produced by *Streptomyces hygroscopicus* and *Streptomyces cyanogriseus*, include milbemycin oxime and moxidectin [14,15]. Both families have a common pharmacophore, a 16-member macrocyclic lactone ring [15]. To exert their anti-parasitic function, ML bind to glutamate-gated chloride channels, causing paralysis in nematodes, insects and arachnids [16]. Due to its broad spectrum and its unique pharmacological and safety profile, ML are widely used in veterinary medicine against endo and ectoparasites in pets and livestock [17]. In humans, ivermectin has been widely used over the last 30 years in mass medication campaigns for the treatment and elimination of human onchocerciasis and lymphatic filariasis [18,19,20] alongside other human diseases [16,19]. Recently, moxidectin was also approved for the treatment of onchocerciasis by the US Food and Drug Administration (FDA) [21]. Researchers continue to explore new applications for ivermectin to reduce malaria, leishmaniasis, and trypanosomiasis transmission by vector control [20,22], or as anti-viral agents against an extensive range of RNA viruses such as HIV-1, Dengue virus, or SARS-CoV-2, among others [23,24]. However, caution is needed when efficacy assessment is based solely on in vitro data [25].

Initially thought to be inactive against bacteria, we previously identified the bactericidal activity of ML against certain mycobacterial species, including *Mycobacterium tuberculosis*, *Mycobacterium bovis*, *Mycobacterium smegmatis*, *M. ulcerans* and *M. marinum*, [26,27]. The aim of this study was to expand previous reports to include a wider range of mycobacterial species and evaluate the in vitro anti-mycobacterial activity of ML against NTM.

## 2. Results

### 2.1. Milbemycin Oxime Was the Most Active Macrocyclic Lactone against NTM with Broad-Spectrum Antimycobacterial Activity

Eight ML were tested against a panel of 80 NTM, including reference strains and clinical isolates from local hospitals (Figure 1, Appendix A). Milbemycin oxime was the most active ML against all the species tested: *M. abscessus*, *M. chelonae,* and *M. fortuitum* showed the highest Minimum Inhibitory Concentration (MIC) in a mean range of 4–16 mg/L; *M. avium,* 2–16 mg/L and the remaining NTM species with MIC values lower than 4 mg/L. Emamectin and selamectin also showed promising activity, although they were less potent or had a narrower activity spectrum than milbemycin oxime. In the case of selamectin, MIC distribution was variable with good activity against most species (MIC ≤ 8 mg/L) but with no in vitro activity against *M. abscessus* and *M. chelonae* (MIC ≥ 32 mg/L). *M. avium* strains displayed a bimodal MIC distribution (MIC = 1–8 mg/L and ≥32 mg/L). Ivermectin, abamectin, doramectin, and moxidectin were only moderately active against some strains, with an overall lack of activity (MIC > 32 mg/L). Eprinomectin showed no in vitro activity (MIC > 32 mg/L).

### 2.2. Milbemycin Oxime Did Not Show any Inducible Resistance against Rapidly Growing Mycobacteria, in Contrast to Clarithromycin

ML are structurally similar to antibacterial macrolides— both contain a macrocyclic lactone ring [15,18,28]. Current CLSI guidelines recommend performing a second MIC measurement after an extended incubation period of at least 14 days in order to detect inducible macrolide resistance in rapidly growing mycobacteria (RGM) [29]. We performed MIC determinations after the standard 3 days and extended 14 days of incubation to determine whether milbemycin oxime displayed a similar inducible resistance pattern as the first-line macrolide antibiotic clarithromycin (Table 1). Consistent with the distribution of the macrolide inducible resistance *erm* genes in the different species, *M. abscessus* sp. *abscessus*, *M. abscessus* sp. *bolletii,* and *M. fortuitum* showed a strong increase in their clarithromycin MIC values after 14 days of incubation compared to values after 3 days; this was not observed for *M. abscessus* sp. *massiliense,* or *M. chelonae* [30,31,32,33]. In contrast, MIC values of milbemycin oxime were the same against different species at both incubation times.

### 2.3. Macrocyclic Lactones Displayed Selective Dose-Dependent Bacteriostatic or Bactericidal Activities against Different NTM Species

To further characterize the in vitro activity of ML against NTM, we performed Time Kill Assays (TKA) of milbemycin oxime (Figure 2), ivermectin and selamectin (Appendix A) at two concentrations against some clinically relevant NTM species; the first-line drug clarithromycin was included as an internal control. TKA of clarithromycin explained the observed increase in MIC values at 3 and 14 days (Table 1), i.e., there was a growth rebound in *M. abscessus* sp. *abscessus*, *M. abscessus* sp *bolletii* and, especially, *M. fortuitum*— probably due to the development of inducible resistance against clarithromycin. In the case of *M. abscessus* sp. *massiliense* and *M. chelonae*, lacking a macrolide inducible *erm* resistance gene, both concentrations of clarithromycin showed rapid bactericidal and sterilizing activity [29,30,31,32,33]. For those species with macrolide inducible resistance, milbemycin oxime 80 mg/L exhibited the best activity, preventing regrowth in *M. abscessus* sp. *bolletii* and *M. fortuitum.* Importantly, milbemycin oxime was equally or slightly more active than clarithromycin against *M. abscessus* sp. *abscessus.* Milbemycin oxime was especially bactericidal against *M. kansasii* with sterilizing activity at both concentrations tested, but bacteriostatic against *M. avium* and *M. intracellulare* (although at 80 mg/L the effect was prolonged, suggesting that re-dosing could increase activity) (Figure 2). The anti-mycobacterial activities of selamectin and ivermectin were also characterized by TKA against RGM, displaying a reduced activity compared to milbemycin oxime. While ivermectin was inactive, selamectin showed some activity against *M. chelonae* and *M. fortuitum* but not against MABSC. Both concentrations of selamectin displayed a bacteriostatic profile against slow-growing mycobacteria (SGM) (Appendix A).

## 3. Discussion

Macrolide antibiotics are the backbone of current NTM therapies, typically administered in combination with amikacin [7,34]. However, the development of macrolide resistance occurs frequently, thus restricting their therapeutic efficacy. Resistance to macrolides monitored in vitro is associated with poorer outcomes [10,13]. Although there have been some recent advances in NTM therapy, there has not been much improvement regarding new drugs or regimens, since the macrolide-based multidrug therapy was established in 1990 [2]; this could be, in part, due to the traditionally reduced interest from pharmaceutical companies in developing specific NTM therapies, together with the lack of both awareness of this neglected problem and robust data defining the global burden of NTM disease [1]. There is thus an urgent need to identify new drugs and develop more effective and safer NTM regimens, especially to treat infections resistant to macrolides. We previously demonstrated the in vitro activity of some ML against mycobacteria, including pathogenic *M. tuberculosis* and *M. ulcerans* [26,27]. Here, we characterized the potential role of ML against NTM infections.

The in vitro antimicrobial activities of eight ML were tested against a panel of 80 NTM strains including nine species (Figure 1, Appendix A). Milbemycin oxime was the most active (MIC ≤ 8 mg/L), with broad-spectrum activity against all species tested. In the case of selamectin (the compound with the best potential based on available pharmacokinetic properties [27]) the MIC distribution was variable, with good activity against most species (MIC ≤ 8 mg/L), but with limited in vitro activity against *M. abscessus* and *M. chelonae* (MIC ≥ 32 mg/L).

A major concern in NTM therapy is the development of macrolide resistance conferred by two well-known genes: *rrl* and *erm* [35]. Mutations in the *rrl* gene providing macrolide resistance rarely emerge during therapy. In contrast, macrolide-inducible *erm*-dependent resistance is common in NTM therapy, with *erm* genes identified in several RGM [30,32,35]. Subspecies of *M. abscessus* differ at the *erm41* locus; in *erm41*, the T polymorphism at nucleotide position 28 (T28) of the structural gene confers inducible resistance, while isolates with the C polymorphism (C28) remain susceptible. *M. abscessus* sp. *bolletii* includes the T28 polymorphism, *M. abscessus* sp. *abscessus* can have either (T28 or C28), and the *erm41* gene in the *M. abscessus* sp. *massiliense* is truncated and non-functional [30]. In *M. chelonae* these mutations have not been observed, while *M. fortuitum* can express the inducible *erm39* isoform [31,32]. CLSI guidelines recommend reporting MIC values after 3 and 14 days of incubation to identify clarithromycin resistance due to an *erm* induction mechanism [29]. In our study, we confirmed these observations: *M. abscessus* sp. *abscessus, M. abscessus* sp. *bolleti,* and *M. fortuitum* showed an MIC increase from day 3 to day 14; in contrast, *M. abscessus* sp. *masiliense* and *M. chelonae* did not show this increase (Table 1). ML are structurally similar to macrolide antibiotics, both containing a macrocyclic lactone ring [15,18,28]; ML are 16-membered macrocyclic lactone ring compounds, while macrolides could have 12-, 14-, 15-, or 16-membered macrocyclic lactone rings [28]. We thus investigated whether resistance to ML could also be inducible in RGM, similar to clarithromycin. We found that the MIC values of milbemycin oxime remained constant at both 3 and 14 days of incubation in all five strains tested, suggesting a lack of an inducible resistant mechanism (Table 1). To further understand the antimicrobial activity of ML, we also performed TKA in parallel with the first line drug clarithromycin. ML displayed a bacteriostatic or bactericidal activity dependent on the NTM species tested. Importantly, a detailed analysis of the time–kill kinetic curves revealed better or similar potency of milbemycin oxime and clarithromycin against some of the strains (Figure 2). In the case of *M. fortuitum*, for which the induction of clarithromycin resistance was more evident (regrowth after only one day of drug exposure), at a high dose (80 mg/L), milbemycin oxime was able to prevent regrowth after 14 days of incubation.

ML are orally active compounds with fast absorption [14], wide distribution in the body, and long residence times [18,19]. Due to their specificity for parasitic targets they have a high margin of safety in mammals at clinical doses [36]. Ivermectin and moxidectin are the only ML approved for human use, ivermectin being the most widely used. At clinical doses, ivermectin is extremely well tolerated and safe [37]. Similarly, moxidectin, recently approved for humans, has been widely used in veterinary medicine. However, in this study, neither of these compounds showed significant activity against any of the mycobacterial strains tested. In contrast, milbemycin oxime, the most active of the ML according to in vitro data, and selamectin, the ML with the best translational potential based on PK considerations [27], displayed significant activity in our study. However, at present, these are commercialized and licensed only for veterinary use (cats and dogs) [17]. Their inclusion in NTM therapy would thus require a repurposing approach that could be facilitated thanks to the extensive pharmacological data available from animal studies.

Milbemycin oxime reaches higher concentrations in blood than ivermectin with an improved safety profile due to reduce P-gp binding [27]. At half (0.25 mg/kg) and double (0.92 mg/kg) the standard dose of milbemycin oxime typically used in dogs, C_max_ values could achieve 79.33 µg/L and 353 µg/L, respectively [27]. These concentrations are ca. 25-fold lower than our observed effective dosing for anti-mycobacterial activity (ca. 8 mg/L); thus, simple extrapolation would approximate a potential effective dose to be about 25 mg/kg, although appropriate projections should be calculated to this end. Previous work has shown a dose ca. 100-fold higher (100 mg/kg) caused severe, adverse, but transient effects in mice while doses of 300 mg/kg produced severe toxicosis and death [38]. In the case of selamectin, oral doses of 24 mg/kg have been tested reaching C_max_ values of 7.6 mg/L [39]; this is within the effective anti-mycobacterial range. In toxicity studies performed in mice, doses of 30 mg/kg of selamectin did not cause any adverse effect, while only mild toxicity signs were observed at 100 mg/kg and 300 mg/kg [38]. Although milbemycin oxime may have some margin for a safe dose increase, without detailed human dose prediction studies, current information suggests that selamectin could be the ML of choice for some NTM infections.

## 4. Materials and Methods

### 4.1. Mycobacterial Strains

Eighty NTM strains (within nine species) were used: *M. abscessus* (n = 10), *M. avium* (n = 15), *M. chelonae* (n = 6), *M. fortuitum* (n = 9), *M. gordonae* (n = 8), *M. intracellulare* (n = 6), *M. kansasii* (n = 18), *M. marinum* (n = 5), and *M. microti* (n = 3). Reference strains were procured from strain collections: from American Type Culture Collection *M. abscessus* sp. *abscessus* ATCC 19977, *M. chelonae* ATCC 19235, *M. fortuitum* ATCC 6841, *M. intracellulare* ATCC 35761, *M. kansasii* ATCC 12478; from *Spanish Type Culture Collection*
*M. avium* CECT 7404; and from Culture Collection University of Gothenburg *M. abscessus* sp. *bolletii* CCUG 50184 and *M. abscessus* sp. *masiliense* CCUG 48898. Clinical strains were provided by the Clinical University Hospital Lozano Blesa, Zaragoza, Spain; by the Veterans Affairs Medical Center, Syracuse, New York; by St George’s University of London, UK; and by the University of British Columbia, Vancouver, Canada. Species were identified using GenoType *Mycobacterium* CM/AS assay (Hain Lifescience GmbH, Nehren, Germany) and PCR-restriction fragment length polymorphism analysis [40].

### 4.2. General Growth Conditions and Drugs

Strains were grown in Middlebrook 7H9 broth (Difco) supplemented with 10% albumin, dextrose, and catalase (ADC) (Difco), and 0.5% glycerol. Middlebrook 7H10 agar plates (Difco) supplemented with 10% oleic acid, albumin, dextrose, and catalase (OADC) (Difco) and 0.2% glycerol were used for bacterial CFU enumeration. All strains were incubated at 35–37 °C. Compounds (and suppliers) were: abamectin, doramectin, clarithromycin, amikacin, and linezolid (Sigma-Aldrich, Steinheim, Germany); emamectin and eprinomectin (LKT Labs, Minnesota, MN, USA); ivermectin (Alpha Diagnostic, Texas, TX, USA); and milbemycin oxime, moxidectin, and selamectin (European Pharmacopeia). Stocks solutions (10 mg/mL) were dissolved in DMSO (except amikacin, which was dissolved in sterile water), aliquoted, and stored at −80 °C until use.

### 4.3. Drug Susceptibility Testing

MIC determinations were performed by broth microdilution assays in a 96-well plate format by serial 2-fold dilutions of test compounds as previously described [26]. Briefly, mycobacteria were added to a final cell density of 5 × 10^5^ cells/mL. Positive and negative growth controls were included in every plate for each strain. Plates were incubated at 35–37 °C for two days (and thirteen days in the case of clarithromycin) for the RGM and five days for the SGM. After the incubation period, the redox indicator MTT (3-(4,5-dimethylthiazol-2-yl)-2,5-diphenyl tetrazolium bromide) was added to the wells and incubated overnight. Then, optical density was read at 580 nm to measure the MTT to formazan conversion, an indicator of bacterial growth. The lowest drug concentration that inhibited conversion by 90% compared to the internal negative control was used to define the MIC value. Experiments were done in duplicate at least three times.

### 4.4. Time Kill Assays

Bacterial cultures were inoculated at a final cell density of 10^5^ cells/mL and a final volume of 10 mL media in 25 cm^2^ tissue culture flasks. Drugs were added at designated concentrations based on previously calculated MIC values. Flasks were then incubated at 35–37 °C for 14 days. At every time point, bacterial suspensions were thoroughly mixed, serially diluted 10-fold in PBS with 0.1% tyloxapol (Sigma-Aldrich), and 100 µL aliquots plated on agar plates. CFUs were enumerated after 3 days of incubation for RGM and 9 days for SGM at 35–37 °C. Plates were checked again for late growers. A growth control (no drug) was always included for each strain. Experiments were performed in duplicate at least twice.

## 5. Conclusions

The surge in NTM infection incidence and poor treatment outcomes highlights an urgent need to identify new therapeutic alternatives. We profiled in vitro activities of several ML against a panel of NTM pathogens, including *M. avium*, *M. kansasii,* and *M. abscessus* that typically caused pulmonary NTM infections. Milbemycin oxime displayed the best antimycobacterial profile. Selamectin also had good activity and could offer a more valid alternative based on its PK properties. Although these compounds are currently only licensed for veterinary use, the extensive pharmacological data available in different animal species could facilitate a repurposing drug development approach.

## Figures and Tables

**Figure 1 antibiotics-10-00381-f001:**
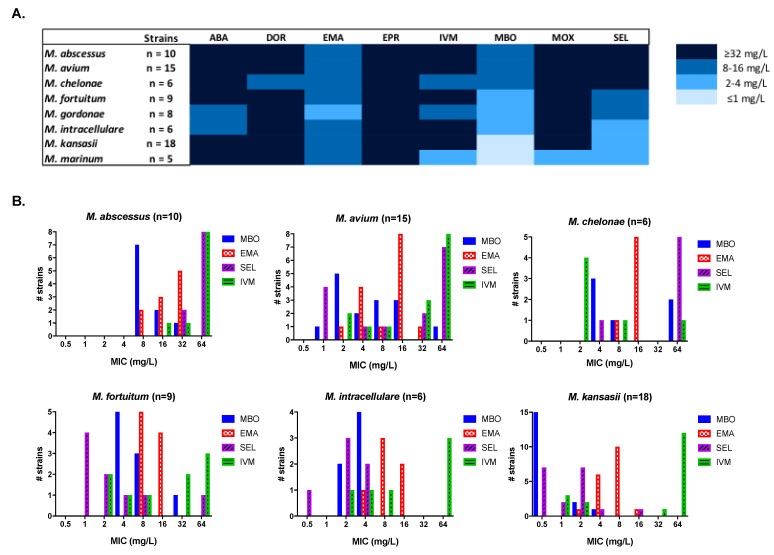
In vitro susceptibility of nontuberculosis mycobacteria (NTM) against different avermectins and milbemycins. (**A**) Heat map representation of the most common MIC values of eight ML against a panel of NTM strains. (**B**) MIC distribution of the most active ML against clinically relevant NTM strains. ABA, abamectin; DOR, doramectin; EMA, emamectin; EPR, eprinomectin; IVM, ivermectin; MBO, milbemycin oxime; MOX, moxidectin; SEL, selamectin.

**Figure 2 antibiotics-10-00381-f002:**
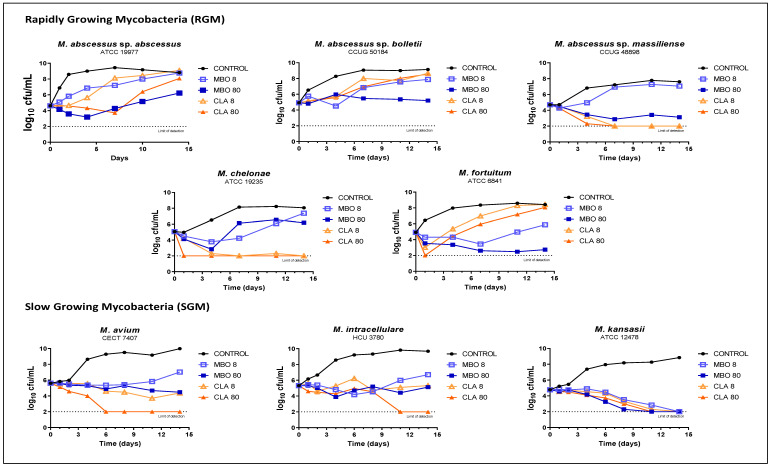
In vitro Time Kill Assays of milbemycin oxime against different RGM and SGM species. Clarithromycin was also included as internal control of activity. Compounds were tested at 8 mg/L and 80 mg/L, i.e., 1×MIC and 10×MIC of milbemycin oxime, respectively. MBO, milbemycin oxime; CLA, clarithromycin.

**Table 1 antibiotics-10-00381-t001:** Differential MIC values of selected compounds against rapidly growing mycobacteria (RGM) after 3 and 14 days of incubation. Amikacin was included as a control antibiotic not containing a macrocyclic lactone ring. MBO, milbemycin oxime; CLA, clarithromycin; AMK, amikacin.

Strains	MIC (mg/L) for RGM Strains								
	MBO		Fold Change	CLA		Fold Change	AMK		Fold Change
	3 Days	14 Days		3 Days	14 Days		3 Days	14 Days	
*M. abscessus* sp. *abscessus* ATCC 19977	8	8	1	4	128	32	16	64	4
*M. abscessus* sp. *bolletii* CCUG 50184	8	8	1	4	128	32	16	32	2
*M. abscessus* sp. *masiliense* CCUG 48898	8	8	1	0.5	2	4	16	64	4
*M. chelonae* ATCC 19235	8	16	2	0.12	0.5	4	nd	nd	nd
*M. fortuitum* ATCC 6841	8	8	1	8	64	8	4	4	1

## Data Availability

The data presented in this study are available in the article or Appendix A.

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
