# Peer review of "Repurposing Avermectins and Milbemycins against Mycobacteroides abscessus and Other Nontuberculous Mycobacteria"

_antibiotics, 2021, doi:10.3390/antibiotics10040381_

Round 1
Reviewer 1 Report
Dear Sirs,
The article entitled “Repurposing avermectins and milbemycins against Mycobacteroides abscessus and other nontuberculous mycobacteria” has the main goal of characterizing anti-mycobacterial activity of macrocyclic lactones against multiple nontuberculous mycobacteria species such as Mycobacteroides abscessus. Content is new and very interesting, easy to read.
Just some simple remarks:
Abstract:
Concrete results of the time kill assays are missing.
Introduction:
Lines 36-37: could you add concrete numbers related to incidence and prevalence?
Line 37, for instance: can the references be added simultaneously rather than each one individually?
Results:
Figure 1. Statistical analysis is needed to reinforce results obtained.
Figure 2. Why weren’t concentrations adjusted to obtain log 0?
Materials and Methods
Lines 259-260: MTT measures cell metabolic activity. To infer on cell growth, you need to quantify cell number or DNA content and normalize the results. If not, it is important to clarify what you have determined.
Author Response
Point 1: Abstract. Concrete results of the time kill assays are missing
Response 1: They have been now included. It reads now:
“(…) For this, MIC values of eight ML were determined against 80 strains belonging to nine different NTM species. Macrocyclic lactones showed variable ranges of anti-mycobacterial activity that were compound and species dependent. Milbemycin oxime was the most active compound displaying broad-spectrum and bactericidal activity with MIC lower than 8 mg/L. Time kill assays confirmed MIC data and showed bactericidal and sterilizing activity of some compounds. (…)”
Point 2: Introduction. Lines 3637: could you add concrete numbers related to incidence and prevalence?
Response 2: Incidence and prevalence of NTM infections both pulmonary and extrapulmonary NTM infections are underestimated and difficult to calculate due to several reasons as discussed in the following papers:
- “Non-tuberculous mycobacterial infections—A neglected and emerging problem. International Journal of Infectious Diseases 92S (2020) S46–S50”
- “NTM Drug Discovery: Status, Gaps and the Way Forward. Drug Discov. Today 2018, 23, 1502–1519”
- “The Rise of Non-Tuberculosis Mycobacterial Lung Disease. Immunol. 2020, 11, 1–12”
As such, since incidence vary depending on the affected population and prevalence depending on continents, regions, and countries it is difficult to provide concrete numbers. The new sentence in the intro to reflect this now reads:
“The incidence of documented infections caused by nontuberculous mycobacteria (NTM) are on the rise worldwide and becoming a new neglected global health concern [1–3]. Reasons include NTM outbreaks in cosmetic and surgical procedures, potential transmission of Mycobacteroides abscessus among patients with cystic fibrosis (CF) and the increasing number of vulnerable individuals at risk of developing these infections, i.e. immunocompromised patients or patients with chronic pulmonary pathologies [4–6]. Even though, the true global burden of NTM disease is vastly under-diagnosed due to several unresolved obstacles including the non-specific presentation of NTM disease and limited diagnostic capacity in many countries leading to under diagnosis [1–3].”
Point 3: Line 37: for instance: can the references be added simultaneously rather than each one individually?
Response 3: References have been modified accordingly in the whole text.
Point 4: Results. Figure 1. Statistical analysis is needed to reinforce results obtained.
Response 4: We would kindly appreciate any clarification on the type of statistical analysis to be performed in here. Figure 1 is a summary of data contained in Table S1 where all the MIC data is provided. Typically, no statistical analysis is performed with MIC data. Fig 1.A is a graphical representation of MIC data contained in Table S1 and Fig 1.B is a representation with the total number of strains displaying each MIC value.
We believe that by providing all MIC raw data in Table S1, the reader has all the necessary tools to evaluate the data presented.
Point 5: Figure 2. Why weren’t concentrations adjusted to obtain log 0?
Response 5: Due to the dilutions and volumes needed for CFU plating this technique has a limit of detection of 2-log10.
Point 6: Materials and Methods. Lines 259-260, MTT measures cell metabolic activity. To infer on cell growth, you need to quantify cell number or DNA content and normalize the results. If not, it is important to clarify what you have determined.
Response 6: The MTT methodology has been widely used as a biomarker surrogate of cell growth by measuring the conversion of MTT to formazan. Information is in reference [26] and those therein.

Reviewer 2 Report
Dear Authors, congratulations for your paper. NTM (nontuberculous mycobacteriosis) they are very difficult for clinicians to treat and represent a little known burden of disease and disability.
It is very difficult in my opinion give some suggetons becuases this paper is already in high quality
Below my suggestions:
- Introduction: well. Any data on global burden of NTM? Furthermore, Line 75-78 add (Malaria and COVID-19: Common and Different Findings. Trop Med Infect Dis. 2020 Sep 6;5(3):141. doi: 10.3390/tropicalmed5030141.
- Methods and results: no suggestions
- Discussion: give some publich and global health proposal to reduce the burden and increase the knoledge on NTM . Ex training duringn medical school, or during specialitation or other proposal that can came from your excellent paper.
Author Response
Point 1: Introduction. Any data on global burden of NTM?
Response 1: See answer (Point 2) to Reviewer #1.
Point 2: Line 7578: add (Malaria and COVID19: Common and Different Findings. Trop Med Infect Dis. 2020 Sep 6;5(3):141. doi: 10.3390/tropicalmed5030141.
Response 2: We appreciate the reviewer suggestion and the opportunity to read an interesting paper on COVID-19 and malaria co-infection, their potential interconnection and its impact in vulnerable populations. However, we felt that this elegantly written viewpoint suggested by reviewer #2 does not fully fit in the referred paragraph, where applications of ivermectin are suggested as a vector control or directly as anti-viral agent. A minor edit has been included to clarify this point. It now reads:
“(…) Researchers continue to explore new applications for ivermectin to reduce malaria, leishmaniasis and trypanosomiasis transmission by vector control [20] [22] or as anti-viral agents against an extensive range of RNA virus such as HIV-1, Dengue virus or SARS-CoV-2, among others [23, 24], although caution is needed when efficacy assessment is based solely on in vitro data [25].”
Point 3: Methods and results: no suggestions
Point 4: Discussion. Give some public and global health proposal to reduce the burden and increase the knowledge on NTM. Ex training during medical school, or during specialitation or other proposal that can came from your excellent paper.
Response 4: We appreciate this kind suggestion but we felt that this interesting issues for discussions belong to other areas of research more related to global health and do not align with the specifics of our discussion which is focused on the potential use of the avermectins as anti-NTM agents from more pharmacological point of view. However, we have added a sentence at the beginning of the introduction to highlight some of the reasons for the current state of the NTM problem. The text now reads as follows:
“Macrolide antibiotics are the backbone of current NTM therapies, typically admin-istered in combination with amikacin [7,34]. However, the development of macrolide resistance occurs frequently, thus restricting their therapeutic efficacy. Resistance to macrolides monitored in vitro is associated with poorer outcomes [10,13]. Although there have been some recent advances in NTM therapy, there has not been much improvement regarding new drugs or regimes since the macrolide-based multidrug therapy was established in 1990 [2]; this could be in part due to the traditional reduce interest from pharmaceutical companies in developing specific NTM therapies, together with the lack of both awareness on this neglected problem and robust data defining the global burden of NTM disease [1]. There is thus an urgent need to identify new drugs and develop more effective and safer NTM regimens, especially to treat infections resistant to macrolides. We previously demonstrated the in vitro activity of some ML against mycobacteria, including pathogenic M. tuberculosis and M. ulcerans [26-27]. Here, we characterized the potential role of ML against NTM infections.”